# Elevated aqueous endothelin-1 concentrations in advanced diabetic retinopathy

Hae Min Kang[1,2]*, Md. Hasanuzzaman[3], So Won Kim[4], Hyoung Jun Koh[4], Sung Chul Lee[5]

1 Quantitative method, Harvard TH Chan School of Public Health, Boston, Massachusetts, United States of America, 2 Catholic Kwandong University College of Medicine, Incheon, Republic of Korea, 3 Department of Pharmacology, Asan Medical Center, University of Ulsan College of Medicine, Seoul, Republic of Korea, 4 Department of Ophthalmology, Yonsei University College of Medicine, Seoul, Republic of Korea, 5 Department of Ophthalmology, Konyang University Hospital, Daejeon, Republic of Korea

* haeminkang@hsph.harvard.edu

**Data Availability Statement:** Data cannot be shared publicly because of the decision of IRB in International St.Mary's Hospital. Because Hae Min Kang, the first and the corresponding author, is the only retinal specialist in the hospital, and all the

## Abstract

### Purpose

Endothelin-1 (ET-1) is a potent vasoconstrictor which seems to be involved in the pathogenesis of diabetic retinopathy (DR). However, studies on intraocular ET-1 in DR have been limited. Thus, we investigated aqueous ET-1 levels in patients with DR.

### Methods

A total 85 subjects were included in this prospective study. Those were classified into three groups: advanced DR group included those with severe nonproliferative DR or proliferative DR, whereas early DR group included those with mild nonproliferative DR or moderative nonproliferative DR. Those who underwent cataract surgery and had no concomitant ocular disease were included in the control group. Aqueous humor levels of ET-1 were obtained before intravitreal bevacizumab injection (IVB) and after 1 month for the DR patients, and at the time of cataract surgery for the control group.

### Results

Advanced DR group included 40 eyes (47.1%), whereas early DR group did 19 eyes (22.4%), and control group (26 eyes, 30.5%). Mean aqueous ET-1 level was 10.1±4.1 pg/mL (6.0–21.0 pg/mL) in advanced DR group, 1.9±0.7 pg/mL (0.6–2.8 pg/mL) in early DR group, and 2.1±1.0 pg/mL (0.7–3.9 pg/mL) in control group (P < 0.001). Advanced DR group was further subdivided into severe nonproliferative DR (15 eyes, 12.8%) and proliferative DR (25 eyes, 34.3%). Mean aqueous ET-1 level was 10.1±4.3 pg/mL (6.0–20.1 pg/mL) in patients with severe nonproliferative DR, and 10.0±4.0 pg/mL (6.0–21.0 pg/mL) in those with proliferative DR (P = 0.928) at baseline. Mean ET-1 level at 1 month after intravitreal injection was 2.5±1.0 pg/mL (0.3–4.8 pg/mL) in patients with severe proliferative DR and 2.9

patients were recruited in the International St. Mary's Hospital. IRB decided that the patients' information can be identifiable even by sex and age with specific diagnosis and treatment. Data are available from the approval of Ethics Committee (contact via Je Hoon Park, ceccil@ish.ac.kr) for researchers who meet the criteria for access to confidential data.

**Funding:** This research is supported by a grant of the Korea Health Technology R & D Project through the Korea Health Industry Development Institute (KHIDI), funded by the Ministry of Health & Welfare, Republic of Korea (Grant number: HI21C1251). None of the authors have a proprietary interest in this work. The sponsor or funding organization had no role in the design or conduct of this research.

**Competing interests:** The authors have declared that no competing interests exist.

±1.7 pg/mL (1.0–7.0 pg/mL) in those with proliferative DR (P = 0.443). Mean aqueous ET-1 level was significantly reduced in both groups (P < 0.001, respectively).

## Conclusion

The mean aqueous ET-1 level was significantly higher in the eyes with advanced DR than those with early DR and the control group. The mean aqueous ET-1 level was significantly reduced after intravitreal injections in the advanced DR group. Based on our results, future studies on the exact role of ET-1 in the pathogenesis of DR and future implication for intervention would be helpful for managing DR.

## Introduction

Diabetic retinopathy (DR) is one of the most common retinal vascular diseases threatening vision in all age groups [1,2]. In Korea, 14% of the population has diabetes mellitus (DM), and 11–19% of those with DM have DR [3–6]. Because DR can lead to permanent vision loss, DR appears to have significant socioeconomic effects, especially when it affects younger, working-age individuals.

Microvascular alterations are key features of DR, including microvascular endothelial cell dysfunction, vascular permeability, increased tissue ischemia, and angiogenesis [7]. Selective loss of capillary pericytes, impairment of retinal vascular autoregulation, and disturbance of capillary circulation are the characteristic retinal vascular changes in early DR [8–10]. Low-grade inflammation due to hyperglycemia subsequently leads to up-regulation of various vaso-active cytokines such as endothelial cell adhesion molecules and vascular cell adhesion molecule-1 [7]. The persistent inflammation and subsequent vascular wall injury lead to up-regulation of vasoconstrictors such as endothelin-1 (ET-1) and vasodilators such as nitric oxide (NO) [11,12]. Because retinal blood flow is auto-regulated primarily by these local vaso-active factors, alteration of the responses to these local factors by pericytes and endothelial cells leads to abnormal vascular tone and alteration of retinal blood flow.

ET-1 is a potent vasoconstrictor peptide, and several studies indicate that ET-1 is involved in the pathogenesis of DR [13–17]. These studies found that the endothelin system, including ET-1, the endothelin system is of importance in mediating retinal changes in diabetes, although mechanisms of the endothelin system alteration as well as their effects might vary among individuals [13–17]. One experimental study suggests that retinal blood flow showed an increased resistivity index, an indicator of vasoconstriction, after 1 month of diabetes [15]. The retina from the diabetic animals showed increased mRNA expression for ET-1, ET-3 and ET A after one month, as well as increased ET B mRNA expression after 6 months [15]. They also found that ET-1 and ET-3 immunoreactivity and endothelin receptor concentrations were increased in the retina of diabetic rats [15]. It has been also shown that 15 minutes after intravitreal injection of ET-1, there were significant prolongation of retinal circulation times and retinal artery constriction in the animal model [16]. However, only a few studies investigated intraocular concentrations of ET-1 in patients with DR, and the results are inconsistent [18–20]. One study investigated vitreous ET-1 levels in patients who underwent pars plana vitrectomy. It found higher elevations in ET-1 levels in patients with proliferative DR (PDR) than in the control group and in those with nonproliferative DR [18]. The patients with nonproliferative DR and macular edema have lower vitreous ET-1 levels than those with proliferative DR [18]. They speculated that the reductions in ET-1 levels in the patients with

nonproliferative DR reflects the increase in retinal blood flow that occurs in nonproliferative DR as it has minimal vasoconstrictive effects [18]. However, this study included a relatively small study population (i.e., 5 patients with proliferative DR and 15 patients with nonproliferative DR) [18]. Another study found significantly lower ET-1 levels in vitreous fluid of patients with PDR than in normal control group patients [19]. Another study found significantly higher levels of vitreous ET-1 in patients with proliferative DR than in those with nonproliferative DR or the control group [20].

In this study, we investigated aqueous ET-1 levels in patients with DR versus a control group. We also evaluated the change in mean aqueous ET-1 level after intravitreal anti-vascular endothelial growth factor (VEGF) injection in patients with advanced DR with macular edema.

## Methods

### Enrollment of study subjects

This prospective, comparative study was performed at the Catholic Kwandong University College of Medicine, International St. Mary's Hospital. The study design and protocol were approved by the Institutional Review Board of International St. Mary's Hospital, Catholic Kwandong University College of Medicine and adhered to the tenets of the Declaration of Helsinki. Written informed consent was obtained from each participant in this study.

Study subjects were chosen as participants from May 2019 to October 2019. The study population was classified into three groups: advanced DR, early DR, and control group. The advanced DR group included the patients with severe non-proliferative DR or proliferative DR. These patients underwent intravitreal anti-VEGF injections for concomitant macular edema on pro re nata basis. At the time of the first intravitreal anti-VEGF injections, two consecutive aqueous samplings right before the injection and 1 month after the first intravitreal anti-VEGF injection were performed to compare if there would be any change in aqueous ET-1 level. For each intravitreal injection, 1.25 mg/0.05 mL of bevacizumab (Avastin®; Genentech Inc., San Francisco, CA, USA) was used. The early DR group included the patients with no gross DR, mild nonproliferative DR, or moderate nonproliferative DR. Classification of the stage of DR in each participant was based on the International Clinical Diabetic Retinopathy and Diabetic Macular Edema Severity Scales [21].

Treatment-naïve patients for DR were included in this study. The control group included the patients who underwent uncomplicated cataract surgery, had no other concomitant ocular disease, and had no history of diabetes mellitus. If both eyes met the inclusion and exclusion criteria, the right eye was chosen for analysis.

The primary outcome measure was comparison of differences in the mean aqueous ET-1 levels of three groups. The secondary outcome measures were the changes in mean aqueous ET-1 levels after the first intravitreal anti-VEGF injections in the patients with advanced DR with macular edema and its clinical effects on the 1-year prognosis.

### Ocular examination

An ophthalmologic examination that included a slit-lamp examination, an intraocular pressure measurement using a non-contact tonometer, and a fundus examination after pupillary dilation was performed at each visit. The refractive error of each eye was measured using an autorefractor, and then converted to spherical equivalents [diopters (D)]. Best-corrected visual acuity (BCVA) of the affected eye(s) was checked using a decimal visual acuity chart; each value was converted to the logarithm of the minimum angle of resolution (logMAR). Spectral domain optical coherence tomography (SD OCT) (Spectralis; Heidelberg Engineering,

Heidelberg, Germany) imaging was performed using enhanced depth imaging. Fluorescein angiography (FA) was performed at the first visit using the Heidelberg Retina Angiograph system (HRA-2; Heidelberg Engineering) with a confocal scanning laser ophthalmoscope. FA was used to evaluate the nonperfusion area and presence of neovascularization.

Central macular thickness (CMT) measured using SD OCT was defined as the mean retinal thickness in the central subfield, a region with a diameter of 1.0 mm around the fovea. The inner and the outer rings had diameters of 3.0 mm and 6.0 mm, respectively. CMT using Early Treatment Diabetic Retinopathy Study (ETDRS) macular grid subfields was automatically calculated using the Heidelberg software. The ETDRS grid was manually centered at the fovea if needed. All SD OCT images were reviewed for segmentation errors.

## Sampling of aqueous humor in study subjects

All ocular procedures were performed by one retinal specialist (HM Kang) using a standardized approach. Each aqueous sampling was routinely performed at the beginning of the ocular procedure in our clinic, which enabled this study. Under sterile operating room conditions, topical anesthesia was achieved using 0.5% proparacaine hydrochloride (Alcain®; Alcon Inc., Fort Worth, TX, USA). A 10% povidone-iodine solution was used for periocular sterilization. The surgeon used a solution of povidone-iodine 10% to clean the patient's skin around the eye and used povidone-iodine 5% for irrigation (Betadine®; Alcon, USA). An eyelid speculum was placed after the drape was applied. After further topical anesthesia using 4% lidocaine, anterior chamber paracentesis was performed to drain 0.1 mL aqueous humor.

## Endothelin-1 concentration measurement

Laboratory procedures for aqueous ET-1 concentration was done as previously decribed in the previous study conducted by previous study [22], and our study on retinal vein occlusion [23]. Aqueous humor samples were immediately stored at -80°C until analysis. On the day of analysis, stored samples were thawed and centrifuged at 400 x g for 5 minutes at 4°C to remove cell contents and cell debris [22,23]. ET-1 was measured using a sandwich ELISA kit (R&D SYSTEMS, QUANTIKINE ELISA endothelin-1 kit, catalog number DET-100) according to the manufacturer's protocol [22,23]. In brief, 75-μL aqueous humor samples were pipetted into 96-well plates pre-coated with a monoclonal antibody specific for human ET-1 and that did not cross-react with other isoforms or species. Same volumes of ET-1 standards (concentrations 25 pg/mL, 12.5 pg/mL, 6.25 pg/mL, 3.13 pg/mL, 1.56 pg/mL, 0.78 pg/mL, 0.39 pg/mL, and 0 pg/mL) were also pipetted into 96-well plates to make a standard curve. Color intensity was measured using a multi-well plate reader (Bio Tek Instruments, Inc., South Korea). The wavelengths for all samples and standards were corrected by subtracting OD540 values from OD450 values. After plotting the standard curve, the corrected OD450 values for samples were extrapolated on the Y-axis and the concentration in pg/mL was measured on the X-axis [22,23].

## Statistical analysis

The data were presented as mean ± standard deviation (range) values. Baseline characteristics included age, sex, refractive error, and BCVA at the initial visit. Medical histories, including hypertension and end-stage renal disease, were obtained from the medical charts of the patients. IBM SPSS Statistics software for Windows, version 22.0 (IBM Corp., Somers, NY, USA) was used for the statistical analyses. For continuous variables, the Kruskal-Wallis test was used for comparisons among three groups, and Mann-Whitney U test was used for comparison between two groups. The chi-square test was used for categorical variables. Repeated-

measures analysis of variance was used to compare values for mean BCVA, mean CMT, and mean aqueous ET-1 levels after intravitreal injections. Pearson's correlation analysis was performed to evaluate for any significant correlations between aqueous ET-1 level with visual and anatomical outcomes in the patients with advanced DR after intravitreal injection. Multiple regression analysis by using stepwise approach was used to investigate the correlation between the baseline factors with the mean aqueous ET-1 level at baseline. Mauchly's test of sphericity and Kolmogorov-Smirnov analyses were used to confirm statistical validity. A P value <0.05 was considered statistically significant.

## Results

### Baseline characteristics of the study population

Eighty-five eyes of 85 patients were included in this study. The advanced DR group included 40 eyes (47.1%), the early DR group included 19 eyes (22.4%), and the control group included 26 eyes (30.5%). The advanced DR group was further subdivided into severe nonproliferative DR (15 eyes, 12.8%) and proliferative DR (25 eyes, 34.3%). The early DR group included no DR (7 eyes, 8.2%), mild nonproliferative DR (10 eyes, 11.8%), and moderate nonproliferative DR (2 eyes, 2.4%). In the overall study population, mean age at the time of diagnosis was 64.4 ±10.9 (37.0–86.0) years. Mean aqueous ET-1 level was 10.1±4.1 pg/mL (6.0–21.0 pg/mL) in the advanced DR group, 1.9±0.7 pg/mL (0.6–2.8 pg/mL) in the early DR group, and 2.1±1.0 pg/mL (0.7–3.9 pg/mL) in the control group (P < 0.001). The results for comparison among the three subgroups are presented in Table 1.

### Changes after intravitreal anti-vascular endothelial growth factor injections for advanced diabetic retinopathy with macular edema

During the 12-month follow-up period, a mean number of 1.7±0.9 (range 1–4) intravitreal bevacizumab injections were performed for the advanced DR group. The mean BCVA value was 0.6±0.4 logMAR (0.1 to 1.3 logMAR) at baseline and 0.3±0.3 logMAR (0 to 1.3 logMAR)

**Table 1. Baseline characteristics of the study population.**

|  | Advanced DR group (N = 40) | Early DR group (N = 19) | Control group (N = 26) | P value |
|---|---|---|---|---|
| Mean age (years) | 58.6±9.6 (37–79) | 69.5±7.9 (52–82) | 69.6±9.7 (51–86) | 0.028* |
| Sex (male) | 14 (35.0%) | 11 (57.9%) | 14 (53.8%) | 0.630† |
| Hypertension | 15 (37.5%) | 7 (36.8%) | 12 (46.2%) | 0.512† |
| ESRD | 8 (20.0%) | 0 | 0 | 0.004† |
| Mean fasting glucose (ml/dl) | 198.2±63.6 (96.0–333.0) | 146.5±39.0 (107.0–245.0) | 111.2±16.7 (87.0–132.0) | <0.001* |
| Mean HbA1c (%) | 7.8±1.3 (6.0–11.0) | 7.8±1.1 (5.8–12.0) | 5.1±0.4 (4.5–5.5) | 0.863* |
| Mean creatinine (ml/dl) | 2.8±3.6 (0.52–15.0) | 0.8±0.4 (0.41–2.17) | 0.7±0.2 (0.37–1.03) | <0.001* |
| Nonperfusion area on FA | 40 (100%) | 0 | N/A | <0.001† |
| NVE on FA | 25 (62.5%) | 0 | N/A | <0.001† |

*The Kruskal-Wallis test was used for continuous variables and

†the chi-square test was used for categorical variables.

Abbreviations: DR, diabetic retinopathy; ESRD, end-stage renal disease.

at 1 month after IVB (P < 0.001). Mean CMT was 415.5±104.5 (352.0 to 810.0 μm) at baseline and 313.0±70.8 μm (205.0 to 483.0 μm) at 1 month after intravitreal anti-VEGF injections (P < 0.001). Mean baseline aqueous ET-1 level was 10.1±4.1 pg/mL (6.0–21.0 pg/mL), which was significantly reduced to 2.8±1.4 pg/mL (0.3–7.0 pg/mL) (P < 0.001) after intravitreal anti-VEGF injections.

At 12 months after the first injection, mean BCVA was 0.3±0.3 logMAR (1.3 to 0 logMAR) and mean CMT was 323.9±103.4 μm (205.0 to 808.0 μm. Mean BCVA and mean CMT values at the 12-month follow-up point were significantly improved, compared with the baseline values (P < 0.001, respectively).

## Comparison between severe nonproliferative diabetic retinopathy and proliferative diabetic retinopathy

Then, we compared various characteristics between the patients with severe nonproliferative DR and those with proliferative DR. Except the mean age, there were no significant differences in various baseline characteristics between two groups. Detailed comparison is shown in Table 2.

There were no significant differences in mean aqueous ET-1 levels, mean BCVAs, and mean CMTs between the severe nonproliferative DR group and the proliferative DR group (P = 0.928, P = 0.315, and P = 0.120, respectively; Figs 1 and 2). After the first intravitreal injection, mean aqueous ET-1 level was significantly reduced in both the patients with severe nonproliferative DR (P < 0.001) and those with proliferative DR (P < 0.001, Fig 1). After 1 month and 12 months after the first intravitreal bevacizumab injection, there were also significant improvement of mean BCVA and mean CMT after the first intravitreal injection in both groups as shown in Fig 2.

## Correlation between aqueous endothelin-1 levels with visual and anatomical outcomes in patients with advanced diabetic retinopathy

We performed correlation analyses between baseline aqueous ET-1 levels and clinical outcomes in the patients with advanced DR. There was no significant correlation between baseline

**Table 2. Comparison between patients with severe nonproliferative diabetic retinopathy (NPDR) versus those with proliferative diabetic retinopathy (PDR).**

| | Severe NPDR (N = 15) | PDR (N = 25) | P value |
|---|---|---|---|
| Mean age (years) | 63.3±8.4 (48–79) | 55.8±9.3 (37–66) | 0.015* |
| Sex (male) | 10 (66.7%) | 14 (56.0%) | 0.372† |
| Hypertension | 5 (33.3%) | 10 (40.0%) | 0.469† |
| ESRD | 1 (6.7%) | 7 (28.0%) | 0.108† |
| Mean fasting glucose (ml/dl) | 198.3±67.5 (96.0–311) | 198.0±70.1 (107.0–333.0) | 0.934* |
| Mean HbA1c (%) | 7.8±0.9 (6.3–9.8) | 7.9±1.2 (6.0–11.0) | 0.956* |
| Mean creatinine (ml/dl) | 1.7±2.3 (0.52–9.6) | 3.4±4.1 (0.6–15.0) | 0.295* |
| Mean number of intravitreal injections | 1.7±0.7 (1–3) | 1.7±1.1 (1–4) | 0.966* |

* Mann-Whitney U tests were used for continuous variables and

†chi-square tests were used for categorical variables.

Abbreviations: BCVA, best-corrected visual acuity; DR, diabetic retinopathy; ESRD, end-stage renal disease.

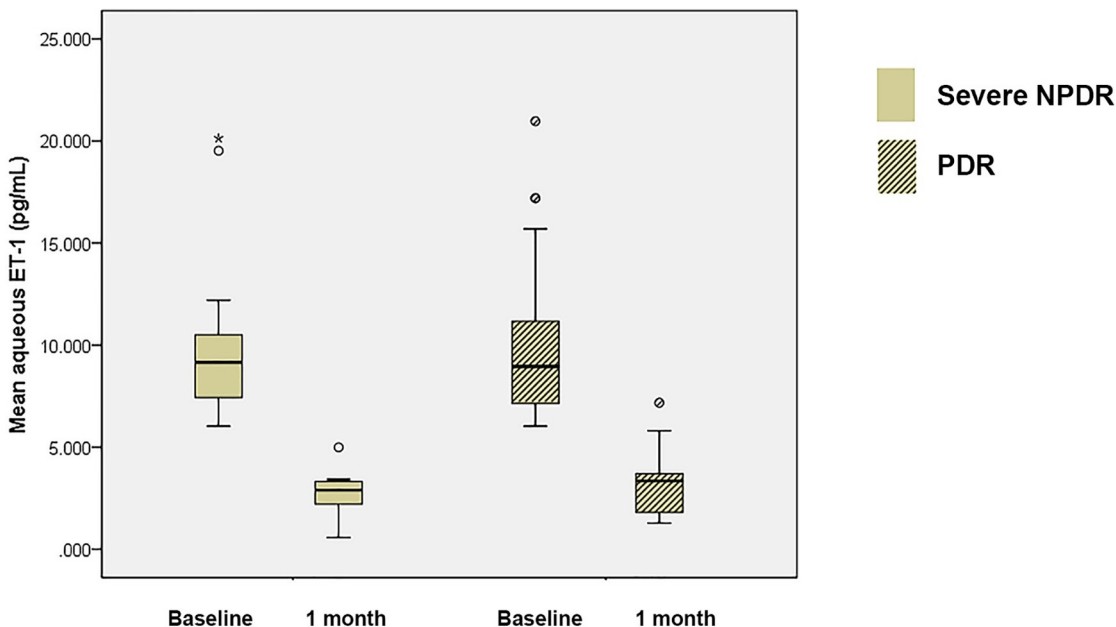

**Fig 1. The changes of mean aqueous endothelin-1 (ET-1) after intravitreal anti-vascular endothelial growth factor (VEGF) injections in the patients with severe nonproliferative diabetic retinopathy (DR) and proliferative DR.** The mean baseline ET-1 level was 10.1±4.3 pg/mL (6.0–20.1 pg/mL) in the patients with severe nonproliferative DR and 10.0±4.0 pg/mL (6.0–21.0 pg/mL) in those with proliferative DR (P = 0.928). Mean aqueous ET-1 level was significantly reduced after intravitreal anti-VEGF injections in both the patients with severe nonproliferative DR (P < 0.001) and those with proliferative DR (P < 0.001).

aqueous ET-1 levels and postoperative visual outcomes at 12 months (P = 0.606). There was also no significant correlation between baseline aqueous ET-1 level and postoperative CMT at 12 months (P = 0.910).

## Possible predictive factors for the baseline aqueous endothelin-1 levels in the study population

We further investigated the possible factors correlated with baseline aqueous ET-1 levels in the study population. Baseline characteristics such as presence of hypertension, chronic renal

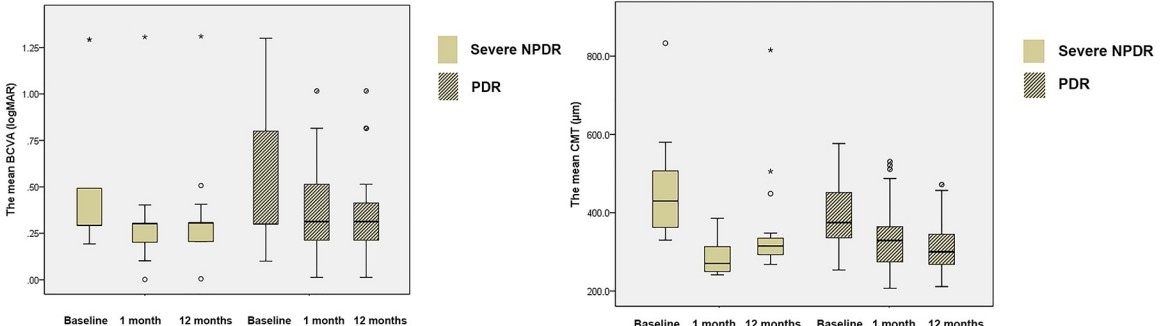

**Fig 2. The changes of mean best-corrected visual acuity (BCVA) and mean central macular thickness (CMT) the patients with severe nonproliferative diabetic retinopathy (DR) and proliferative DR.** (A) The mean BCVA was 0.5±0.4 (1.3~0.2) logMAR in the severe nonproliferative DR group, and 0.6±0.4 (1.3~0.1) logMAR in the proliferative DR group (P = 0.315). Mean BCVA was significantly improved at 1 month and 12 months after intravitreal injection, compared with baseline (P < 0.001 for both). (B) Mean CMT was also significantly reduced at 1 month and 12 months after intravitreal injection, compared with baseline (P < 0.001 for both). At each follow-up point, the mean BCVA (P = 0.667 at 1 month, P = 0.907 at 12 months) and the mean CMT (P = 0.432 at 1 month, P = 0.182 at 12 months) were not significantly different between the patients with severe nonproliferative DR and proliferative DR.

**Table 3. The possible factors associated with the mean aqueous endothelin-1 level in the study population by multiple regression analysis.**

| Characteristics | β | P value |
|---|---|---|
| Age | -0.012 | 0.918 |
| Sex | -0.208 | 0.290 |
| Fasting glucose (mg/dl) | -0.020 | 0.843 |
| Creatinine (mg/dl) | -0.045 | 0.665 |
| Presence of hypertension | -0.018 | 0.581 |
| Presence of end-stage renal disease | -0.135 | 0.194 |
| HbA1c (%) | 0.52 | 0.589 |
| Severity of diabetic retinopathy | 0.691 | <0.001 |

P value < 0.001 is set for statistical significance.

disease/end-stage renal disease on dialysis, age, sex, fasting glucose level, HbA1c, and creatinine level were included. In addition, baseline BCVA, baseline CMT, and severity of DR (early DR and advanced DR) were also included, as shown in Table 3. Among these various factors, the severity of DR was significantly associated with the mean aqueous ET-1 level at baseline (B = 2.116, β = 0.691, P < 0.001; Table 3).

## Discussion

This study is the first to investigate aqueous ET-1 levels in patients with DR. The mean aqueous ET-1 level was significantly higher in the patients with advanced DR than in those with early DR and the control group. The mean aqueous ET-1 level was significantly reduced after intravitreal anti-VEGF injections for macular edema associated with advanced DR, with concomitant improvements in vision and CMT.

A few studies have investigated vitreous ET-1 levels in patients with DR; however, the results are inconsistent [18–20]. We speculate that the different results in intraocular ET-1 levels in these patients with advanced DR [18–20], including in our study, might be due to differences in patient status at the time of study enrollment. In those who undergo pars plana vitrectomy due to advanced DR may have prolonged, progressed disease status or different features than those with treatment-naïve patients such as those in our study. There is another possibility regarding the different intraocular ET-1 levels among the previous studies [18–20] versus our results. The intraocular ET-1 level may be different during each stage of DR, just as suggested in the previous study [15]. As our study and previous studies [13–20], ET-1 seems to involved in the pathogenesis of DR. Along with hypoxic insults in the pathogenesis of DR, increasing endothelin family including ET-1 may aggravate DR itself by constricting retinal vessels and delaying blood circulation time throughout retinal vascular system [13–20]. However, further studies should be followed to identify the exact role of ET-1 in the pathogenesis of DR, as well as possible interactions with diabetes itself and other cytokines involved in the pathogenesis of DR.

In our study, intravitreal bevacizumab injections significantly reduced the mean aqueous ET-1 level. Hypoxia increases hypoxia-inducible factor-1 alpha, leading to upregulation of genes such as erythropoietin, VEGF, and ET-1 [24]. Reduction in VEGF normally leads to reduction in ET-1; this relationship is explained by the stimulatory interaction between VEGF and ET-1 [25]. We speculate that intervention by using intravitreal anti-VEGF agent may reduce abnormally upregulated vasoactive and/or inflammatory cytokines, leading to reduced expression of ET-1. Timely intervention may prevent further vasoconstriction and protect

blood-retinal barrier from further damage in patients with advanced DR. Similarly, a previous study found significant reductions in plasma VEGF, ET-1, and NO levels after pan-retinal photocoagulation in 40 patients with PDR [13]. We could not verify the exact mechanism and further impact of reduced ET-1 level by anti-VEGF agent in this current study because of relatively small study population. However, our results seem to suggest further research point regarding ET-1 level and the role of anti-VEGF agent. Further investigations on the clinical implication of reducing intraocular ET-1 level by anti-VEGF agents can help us understanding the role of ET-1 in DR.

We also investigated the possible factors associated with baseline aqueous ET-1 level in the patients with DR. Among various factors, the severity of DR at baseline was the only factor significantly associated with baseline aqueous ET-1 level among the patients with DM. Our results also support the previous findings that ET-1 is involved in the pathogenesis of DR.[13-20] We also found that there were individual differences of aqueous ET-1 levels among the patients with advanced DR. The range of aqueous ET-1 levels in the patients with advanced DR was from 6.0 to 21.0 pg/mL. We could speculate that there may be other factors affecting aqueous ET-1 levels in the patients with DM. One possible factor can be plasma ET-1 level. Several studies have suggested that plasma ET-1 levels may be transmitted through the peripapillary vasculature [26–28], elevated plasma ET-1 may also contribute intraocular ET-1 levels in patients with DR. In our study, we found that the group of patients with advanced DR had significantly higher prevalence of end-stage renal disease on hemodialysis and older mean age when compared with those with early DR. Plasma ET-1 levels can increase due to mid repeated ischemia-reperfusion events somewhere in the body, concomitant micro- and/or macrovascular complications of DM may also affect plasma ET-1 level [29–31]. Resultant difference in plasma ET-1 levels among individuals with DM may contribute the difference in intraocular ET-1 levels. However, we did not investigate plasma ET-1 levels in this current study, so that we could not confirm the possible relationship between plasma ET-1 and aqueous ET-1. In addition, there may be other unknown factors contributing aqueous ET-1 level. We suggest further researches on the possible factors contributing individual variability of aqueous ET-1 level in the patients with DM, and these can enhance our understanding the role of ET-1 in the pathogenesis of DM.

This study had some limitations, including the relatively small number of patients in the study population. We lacked aqueous humor sampling of the eyes with advanced DR, but without ME, and of plasma ET-1 levels in the study subjects, because of its retrospective manner. In addition, the changes of aqueous ET-1 level during follow-up period may widen our understandings of ET-1 in the pathogenesis and treatment response in advanced DR. Further prospective study which overcomes these weaknesses is needed.

In conclusion, the mean aqueous humor ET-1 level was significantly higher in patients with advanced DR than in those with early DR and the control group. The severity of DR was significantly associated with mean aqueous ET-1 level at baseline in the patients with DM. Further investigations on the exact role of ET-1 in the pathogenesis of DR and its clinical implication can improve our understanding for DR, and may also contribute to the management of these patients.

## Acknowledgments

A. Contributions of Authors: Designed and conducted study (HM Kang, HJ Koh, and SC Lee); collected the data (HM Kang, M Hasanuzzama, and SW Kim); managed, analyzed, and interpreted data (HM Kang, M Hasanuzzama, SW Kim, HJ Koh, and SC Lee); prepared, reviewed, and approved manuscript (HM Kang, M Hasanuzzama, SW Kim, HJ Koh, and SC Lee).

B. Conflict of Interest: The authors declare no competing interests.

C. Other Acknowledgments: None.

## Author Contributions

**Conceptualization:** Hae Min Kang.

**Data curation:** Hae Min Kang, Md. Hasanuzzaman, So Won Kim.

**Formal analysis:** Hae Min Kang, Md. Hasanuzzaman, So Won Kim, Hyoung Jun Koh, Sung Chul Lee.

**Funding acquisition:** Hae Min Kang.

**Investigation:** Hae Min Kang, Md. Hasanuzzaman, So Won Kim, Hyoung Jun Koh.

**Methodology:** Hae Min Kang, So Won Kim, Sung Chul Lee.

**Project administration:** Hae Min Kang, Hyoung Jun Koh, Sung Chul Lee.

**Resources:** Hae Min Kang, Md. Hasanuzzaman, So Won Kim.

**Software:** Hae Min Kang.

**Supervision:** Hae Min Kang, Hyoung Jun Koh, Sung Chul Lee.

**Validation:** Hae Min Kang, Md. Hasanuzzaman, So Won Kim, Hyoung Jun Koh, Sung Chul Lee.

**Visualization:** Hae Min Kang.

**Writing – original draft:** Hae Min Kang, Md. Hasanuzzaman, So Won Kim, Hyoung Jun Koh, Sung Chul Lee.

**Writing – review & editing:** Hae Min Kang, Hyoung Jun Koh, Sung Chul Lee.

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
