## [Decision Letter · Decision Letter 0]

8 Mar 2022

PONE-D-21-28261

Elevated aqueous endothelin-1 concentrations in advanced diabetic retinopathy

PLOS ONE

Dear Dr. Kang,

Thank you for submitting your manuscript to PLOS ONE. After careful consideration, we feel that it has merit but does not fully meet PLOS ONE’s publication criteria as it currently stands. Therefore, we invite you to submit a revised version of the manuscript that addresses the points raised during the review process.

We look forward to receiving your revised manuscript.

Kind regards,

Vikas Khetan, MD

Academic Editor

PLOS ONE

“This research is supported by a grant of the Korea Health Technology R & D Project through the Korea Health Industry Development Institute (KHIDI), funded by the Ministry of Health & Welfare, Republic of Korea (Grant number: HI21C1251).”

“This research is supported by a grant of the Korea Health Technology R & D Project through the Korea Health Industry Development Institute (KHIDI), funded by the Ministry of Health & Welfare, Republic of Korea (Grant number: HI21C1251). None of the authors have a proprietary interest in this work. The sponsor or funding organization had no role in the design or conduct of this research.”

5. We noticed you have some minor occurrence of overlapping text with the following previous publication(s), which needs to be addressed:

- https://journals.plos.org/plosone/article?id=10.1371/journal.pone.0252530

In your revision ensure you cite all your sources (including your own works), and quote or rephrase any duplicated text outside the methods section. Further consideration is dependent on these concerns being addressed.

Reviewers' comments:

Reviewer's Responses to Questions

**Comments to the Author**

1. Is the manuscript technically sound, and do the data support the conclusions?

Reviewer #1: Yes

Reviewer #2: Partly

2. Has the statistical analysis been performed appropriately and rigorously? 

Reviewer #1: Yes

Reviewer #2: Yes

3. Have the authors made all data underlying the findings in their manuscript fully available?

Reviewer #1: Yes

Reviewer #2: Yes

4. Is the manuscript presented in an intelligible fashion and written in standard English?

Reviewer #1: Yes

Reviewer #2: Yes

5. Review Comments to the Author

Reviewer #1: Dear author, I congratulate you for this excellent study. Although it has been known that endothelin-1 levels are upregulated in hypoxic states with concomitant vasoconstriction, this study conclusively shows that the levels are higher with severe grades of DR indicating extreme levels of hypoxia, which reduced after the intravitreal injection of Anti-VEGF agents.

However, please check the manuscript for grammatical errors.

Also, the abstract needs rephrasing: purpose section mentions this as a retrospective study, and the methods section mentions this as prospective.

I also suggest if you could revise the keywords. Having macular edema in this does not make sense. However, I would like to add aqueous endothelin-1, bevacizumab etc.

Overall, an excellent study.

Thank you

Reviewer #2: Authors present a novel work. Following suggestions would help improving the manuscript.

Major Comments

1. Purpose section of Abstract needs to provide a statement about the rationale of the study.

2. Definition of early diabetic retinopathy and advanced diabetic retinopathy needs to be provided in Methods section in Abstract

3. Conclusion in Abstract is just the reiteration of the data in the Results. Conclusion should provide interpretable message of the study along with possible future implications if any.

4. “These studies found that ET-1 has an important role in the pathogenesis and progression of DR.” The role of endothelin 1 in pathogenesis of diabetic retinopathy as indicated in this sentence in Introduction needs to be further elaborated.

5. How was proliferative diabetic retinopathy diagnosed?

6. What is the meaning of “two consecutive intravitreal injections” and how was this number achieved at?

7. “The early DR group included the patients with no gross DR,………..” What is the meaning of “gross DR”?

8. Results is lengthy and needs to be shortened. It discusses the data and provides comparisons. It should include only data and tables. There should not be any discussion of the data; which is part of Discussion section.

9. “A few studies have investigated vitreous ET-1 levels in patients with DR; however,

the results are inconsistent.18-20” Do authors believe that measuring endothelin level in vitreous would have been more appropriate than aqueous levels?

10. “This ‘certain point’ may exist between moderate and severe nonproliferative DR and/or development of macular edema, based on our study results. Advancement to severe nonproliferative DR and concomitant development of macular edema both may be involved in the increase of ET-1 inside the eye” Reviewer believes that this assertion would require different study design where same set of patients are followed up over a period of time to assess endothelin 1 levels alongside evolution of diabetic retinopathy. Or it could be concluded with present study design too, had the sample size been larger enough. Hence the present study should possibly refrain from providing a continuum of endothelin 1 levels alongside evolution of diabetic retinopathy and leave it to future studies.

11. A lot has been discussed about plasma endothelin 1 levels. Reviewer feels that this is speculative and beyond the scope of present study.

12. “If we can detect the critical ‘point’ in patients with DR, interventions antagonizing ET-1 may prevent further disease progression. Then we may prevent further vision loss in these patients with DR.” As stated in previous comment, present study design is not suitable for this assertion.

Minor Comments

1. “For the control group, those who underwent uncomplicated cataract surgery and had no other concomitant ocular diseases were included” This information has been written twice in Methods section.

2. “After reaching ‘certain threshold point’ and progression to advanced DR, then intraocular ET-1 level may be various with different in accordance with each eye status.” This sentence in Discussion needs to be reframed.

6. PLOS authors have the option to publish the peer review history of their article (what does this mean?). If published, this will include your full peer review and any attached files.

Reviewer #1: No

Reviewer #2: No

---

## [Author Response · Author response to Decision Letter 0]

30 Mar 2022

Thank you for the review for PONE-D-21-28261 “Elevated aqueous endothelin-1 concentrations in advanced diabetic retinopathy”, and giving the authors the chance to improve this article. We authors made our every efforts to improve this manuscript according to the comments, and hope that it really improved the quality of this manuscript for PLOS ONE. 

General requirements 

Answer: We authors reviewed the style requirements and the current manuscript, and ensured that this manuscript followed the requirements. 

“This research is supported by a grant of the Korea Health Technology R & D Project through the Korea Health Industry Development Institute (KHIDI), funded by the Ministry of Health & Welfare, Republic of Korea (Grant number: HI21C1251).”

“This research is supported by a grant of the Korea Health Technology R & D Project through the Korea Health Industry Development Institute (KHIDI), funded by the Ministry of Health & Welfare, Republic of Korea (Grant number: HI21C1251). None of the authors have a proprietary interest in this work. The sponsor or funding organization had no role in the design or conduct of this research.”

Answer: Thank you for the comment, and we removed the funding statement from the Acknowledgments section, and rephrased in the Funding Statement section in online submission system. 

Answer: Thank you for the comment. We authors fully understand the data availability of PLOS ONE. However, we also had to follow the decision of the Institutional Review Board of International St. Mary’s Hospital, Catholic Kwandong University College of Medicine. Because Haemin Kang, the corresponding author, is the sole retinal specialist at the time of this study, so the IRB committee worried about any potential of identifying patient information, even by sex and age with specific diagnosis and treatment. Instead, the IRB committee is willing to provide the fully anonymized dataset upon request of investigators. If there is any request for the data, the IRB committee would undergo further discussion, and if possible, it would provide data. The investigators can reach to Prof. Jehoon Park, the head of IRB committee (via ceccil@ish.ac.kr) to request the data used in this study. 

Answer: Thank you for the comment, and in the first paragraph of the Method section, this is written as “This prospective, comparative study was performed at the Catholic Kwandong University College of Medicine, International St. Mary’s Hospital. The study design and protocol were approved by the Institutional Review Board of International St. Mary’s Hospital, Catholic Kwandong University College of Medicine and adhered to the tenets of the Declaration of Helsinki. Written informed consent was obtained from each participant in this study.”

5. We noticed you have some minor occurrence of overlapping text with the following previous publication(s), which needs to be addressed:

- https://journals.plos.org/plosone/article?id=10.1371/journal.pone.0252530

In your revision ensure you cite all your sources (including your own works), and quote or rephrase any duplicated text outside the methods section. Further consideration is dependent on these concerns being addressed.

Answer: Thank you for the comment, and we added this citation to the revised manuscript in the Method section. 

Review Comments to the Author

Reviewer #1: Dear author, I congratulate you for this excellent study. Although it has been known that endothelin-1 levels are upregulated in hypoxic states with concomitant vasoconstriction, this study conclusively shows that the levels are higher with severe grades of DR indicating extreme levels of hypoxia, which reduced after the intravitreal injection of Anti-VEGF agents.

However, please check the manuscript for grammatical errors.

Also, the abstract needs rephrasing: purpose section mentions this as a retrospective study, and the methods section mentions this as prospective.

I also suggest if you could revise the keywords. Having macular edema in this does not make sense. However, I would like to add aqueous endothelin-1, bevacizumab etc.

Overall, an excellent study.

Thank you

Answer: Thank you for the comments. As the reviewer’s recommendation, we authors corrected the abstract and grammatical errors throughout the manuscript. I also revised the keywords, including aqueous endothelin-1 and bevacizumab, omitting macular edema. 

Reviewer #2: Authors present a novel work. Following suggestions would help improving the manuscript.

Major Comments

1.Purpose section of Abstract needs to provide a statement about the rationale of the study.

Answer: Thank you for the comment, and we added a statement about the rationale of the study to the purpose section of Abstracts as following: “Endothelin-1 (ET-1) is a potent vasoconstrictor which seems to be involved in the pathogenesis of diabetic retinopathy (DR). However, studies on intraocular ET-1 in DR have been limited.” 

2. Definition of early diabetic retinopathy and advanced diabetic retinopathy needs to be provided in Methods section in Abstract. 

Answer: Thank you for the comment, and we authors provided the definitions in the Abstract as following: “Those were classified into three groups: advanced DR group included those with severe nonproliferative DR or proliferative DR, whereas early DR group included those with mild nonproliferative DR or moderative nonproliferative DR.”

3. Conclusion in Abstract is just the reiteration of the data in the Results. Conclusion should provide interpretable message of the study along with possible future implications if any.

Answer: Thank you for the comment, and we revised the Conclusion in Abstract, as following “Based on our results, future studies on the exact role of ET-1 in the pathogenesis of DR and future implication for intervention would be helpful for managing DR.”

4. “These studies found that ET-1 has an important role in the pathogenesis and progression of DR.” The role of endothelin 1 in pathogenesis of diabetic retinopathy as indicated in this sentence in Introduction needs to be further elaborated.

Answer: Thank you for the comment, and we added further details in the Introduction section as commented. 

5. How was proliferative diabetic retinopathy diagnosed?

Answer: Thank you for the comment. We authors usually previously diagnosed DR according to fundus findings by using the ETDRS criteria combined with FA and OCT. For this study, we adopted the Disease Severity Scales for Diabetic Retinopathy and Diabetic Macular Edema. We added this to the Method section. 

6. What is the meaning of “two consecutive intravitreal injections” and how was this number achieved at?

Answer: Thank you for the comment, and we authors apologize for this confusion. We authors treat patients with diabetic macular edema on pro re nata basis in this study, and this sentence was meant to be ‘two consecutive aqueous samplings right before and 1 month after the first intravitreal anti-VEGF injection’. We revised the Method section as following: “At the time of the first intravitreal anti-VEGF injections, two consecutive aqueous samplings right before the injection and 1 month after the first intravitreal anti-VEGF injection were performed to compare if there would be any change in aqueous ET-1 level.” Thank you for the comment. 

7. “The early DR group included the patients with no gross DR,………..” What is the meaning of “gross DR”?

Answer: Thank you for the comment. As we previously answered to the comment 5, we used the Disease Severity Scales for Diabetic Retinopathy and Diabetic Macular Edema, and no apparent DR is no visible abnormalities in fundus in the presence of diabetes mellitus. 

8. Results is lengthy and needs to be shortened. It discusses the data and provides comparisons. It should include only data and tables. There should not be any discussion of the data; which is part of Discussion section.

Answer: Thank you for the comment, and as the comment, we revised the Results section much more briefly. 

9. “A few studies have investigated vitreous ET-1 levels in patients with DR; however,

the results are inconsistent.18-20” Do authors believe that measuring endothelin level in vitreous would have been more appropriate than aqueous levels?

Answer: Thank you for the comment. Vitreous fluid can be more directly affected by retinal status, however, those changes are usually reflected to aqueous humor. Endothelin-1 level in vitreous fluid may be more direct indicator for retinal changes, however, vitreous sampling may be much difficult to perform when compared with aqueous sampling. When we consider all pros and cons, we authors think that aqueous can be used as good method to investigate intraocular changes. 

10. “This ‘certain point’ may exist between moderate and severe nonproliferative DR and/or development of macular edema, based on our study results. Advancement to severe nonproliferative DR and concomitant development of macular edema both may be involved in the increase of ET-1 inside the eye” Reviewer believes that this assertion would require different study design where same set of patients are followed up over a period of time to assess endothelin 1 levels alongside evolution of diabetic retinopathy. Or it could be concluded with present study design too, had the sample size been larger enough. Hence the present study should possibly refrain from providing a continuum of endothelin 1 levels alongside evolution of diabetic retinopathy and leave it to future studies.

Answer: Thank you for the comment, and we authors discussed this in further details. We authors agree that we have several limitations to suggest this, and we revised Discussion section as recommended. 

11. A lot has been discussed about plasma endothelin 1 levels. Reviewer feels that this is speculative and beyond the scope of present study.

Answer: We appreciate this comment, and we authors discussed this point. We authors agreed, and re revised this paragraph. 

12. “If we can detect the critical ‘point’ in patients with DR, interventions antagonizing ET-1 may prevent further disease progression. Then we may prevent further vision loss in these patients with DR.” As stated in previous comment, present study design is not suitable for this assertion.

Answer: Thank you for the comment, and we also revised this as recommendation. 

Minor Comments

1.“For the control group, those who underwent uncomplicated cataract surgery and had no other concomitant ocular diseases were included” This information has been written twice in Methods section.

Answer: Thank you for the comment, and we authors apologize this error. We corrected this error. 

2. “After reaching ‘certain threshold point’ and progression to advanced DR, then intraocular ET-1 level may be various with different in accordance with each eye status.” This sentence in Discussion needs to be reframed.

Answer: Thank you for the comment, and we reframed this sentence as recommended. 

Again, we authors really appreciate this chance to improve our manuscript, and if you have any further questions or request, we’ll be glad to hear that. Thank you for all of your efforts.

---

## [Decision Letter · Decision Letter 1]

28 Apr 2022

Elevated aqueous endothelin-1 concentrations in advanced diabetic retinopathy

PONE-D-21-28261R1

Dear Dr. Kang,

We’re pleased to inform you that your manuscript has been judged scientifically suitable for publication and will be formally accepted for publication once it meets all outstanding technical requirements.

Kind regards,

Tomislav Bulum

Academic Editor

PLOS ONE

**Comments to the Author**

1. If the authors have adequately addressed your comments raised in a previous round of review and you feel that this manuscript is now acceptable for publication, you may indicate that here to bypass the “Comments to the Author” section, enter your conflict of interest statement in the “Confidential to Editor” section, and submit your "Accept" recommendation.

Reviewer #1: All comments have been addressed

Reviewer #2: All comments have been addressed

2. Is the manuscript technically sound, and do the data support the conclusions?

Reviewer #1: Yes

Reviewer #2: Yes

3. Has the statistical analysis been performed appropriately and rigorously? 

Reviewer #1: Yes

Reviewer #2: Yes

4. Have the authors made all data underlying the findings in their manuscript fully available?

Reviewer #1: Yes

Reviewer #2: Yes

5. Is the manuscript presented in an intelligible fashion and written in standard English?

Reviewer #1: Yes

Reviewer #2: Yes

6. Review Comments to the Author

Reviewer #1: Thank you for the revised manuscript. All the revisions are acceptable. I do not recommend any further revisions.

Reviewer #2: Authors have modified the manuscript as suggested. There are no more suggestions or modifications for the manuscript.

7. PLOS authors have the option to publish the peer review history of their article (what does this mean?). If published, this will include your full peer review and any attached files.

Reviewer #1: No

Reviewer #2: **Yes: **Kumar Saurabh

---

## [Editor Report · Acceptance letter]

2 May 2022

PONE-D-21-28261R1 

Elevated aqueous endothelin-1 concentrations in advanced diabetic retinopathy 

Dear Dr. Kang:

I'm pleased to inform you that your manuscript has been deemed suitable for publication in PLOS ONE. Congratulations! Your manuscript is now with our production department. 

Kind regards, 

on behalf of

Dr. Tomislav Bulum 

Academic Editor

PLOS ONE